# Verification and Application of Sequence Stratigraphy to Reservoir Characterization of Horn River Basin, Canada

**Juhwan Woo [1,\*] , Jiyoung Choi [2] , Seok Hoon Yoon [3] and Chul Woo Rhee [1,\*]**

1   Department of Earth and Environmental Sciences, Chungbuk National University, Cheongju 28644, Korea
2   Petroleum and Marine Research Division, Korea Institute of Geoscience and Mineral Resources (KIGAM), Daejeon 34132, Korea; jychoi@kigam.re.kr
3   Department of Earth and Marine Sciences, Jeju National University, Jeju 63243, Korea; shyoon@jejunu.ac.kr
\*   Correspondence: jhonta@chungbuk.ac.kr (J.W.); gloryees@chungbuk.ac.kr (C.W.R.); Tel.: +82-43-261-3136 (J.W.)

**Abstract:** Shale reservoirs, the most important unconventional resource, are difficult to characterize. Shale formations require detailed interpretation of geological, petrophysical, and geochemical analyses, and an integration of these disciplines. In terms of geological interpretation, the commonly used sequence stratigraphy analysis includes a lithofacies analysis. The application of sequence stratigraphy to shales facilitates the ability to relate between lithofacies and mineral composition, petrophysical parameters, and kerogen contents, which are affected by depositional setting. The classification of lithofacies is indispensable for reservoir quality prediction. In this study, porosity, permeability, and TOC content largely depend on lithofacies, and their correlation coefficient is relatively high. The sequence stratigraphic interpretation shows that organic carbon content usually increases within the maximum flooding surfaces and decreases stepwise. However, the relationship between total organic carbon contents and systems tract is less direct and redox dependent.

**Keywords:** shale sequence stratigraphy; marine shale; Horn River Formation; Horn River Basin

## 1. Introduction

Shales are a common source of hydrocarbons and can act as reservoirs of oil and gas deposits. The reservoir quality of the shales is complex due to their heterogeneity, which is related to the lithofacies and depositional environments [1,2]. Understanding shale reservoirs can be assessed in the framework of sequence stratigraphy, through integrating diverse data of geological, geophysical, and geochemical features. It helps to correlate and explain the relationship between lithofacies and their geological character, because porosity, permeability, total organic carbon (TOC) content, and mineral composition of shale reflect spatiotemporal variation of fine-grained deposits. Most studies have recognized reservoir quality in relation to relative sea-level fluctuation within a sequence stratigraphic analysis [3–6]. Despite their overall importance, the relationship between reservoir quality and lithology is poorly studied.

The Horn River Formation, deposited in the Western Canada Sedimentary Basin, is one of the well-studied mudstone deposits in Canada. It is composed of an alternating sequence of siliceous mudstone, argillaceous mudstone, and calcareous mudstone [7–10]. Those are divided into the Evie Member, the Otter Park Member and the Muskwa Formation, in ascending order. The geological characteristics of the Horn River Formation have been studied based on sedimentological descriptions, geochemical analysis, wireline log analysis, and sequence stratigraphic analysis [8,10–12]. There have been several sequence stratigraphic interpretations of the Horn River Formation [8,10–12]. The Evie Member was interpreted as deposited during a second-order highstand stage [8,10], and consists of a third-order transgressive systems tract (TST) and an overlying regressive systems tract (RST) [11]. The Otter Park Member is interpreted as a second-order lowstand stage in the middle Devonian carbonate, and a transgressive stage in its upper part [8,10], consisting

of a third-order T–R cycle [11]. The overlying Muskwa Formation formed during the following second-order transgressive stage [8,10], accompanying a third-order TST and an overlying RST [11]. Existing sequence stratigraphic interpretations were worked out by integrating sedimentology, ichnology, and geochemistry, which are marked by a gradual increase in clay contents above the basal surface of forced regression [12]. Previous study defined depositional sequence stratigraphy, including the falling stage systems tract [12].

Sequence stratigraphy analysis is the chronostratigraphic division of sedimentary strata into time-equivalent, genetically related units, with distinct stacking patterns [13,14]. This method has become increasingly important in reservoir and shale gas reservoir characterization [15,16]. This study focuses on lithofacies description, sequence stratigraphy analysis, and petrophysical characterization of the sweet spot from the core and wireline log data.

## 2. Geological Overview

The Horn River Formation was deposited in the Western Canada Sedimentary Basin during the middle to late Devonian (Givetian–Frasnian) period [17,18]. This shale is chronostratigraphically equivalent to the Besa River Formation in the Liard Basin, the Canol Formation in Yukon, and the Duvernay Formation in Alberta [19,20] (Figure 1). The Horn River Formation is overlain by the Fort Simpson Formation, which is poor in organic matter [18]. The southern part of the Horn River Formation was deposited proximal to the paleo-shoreline deposits, whereas the northern part was deposited relatively distal to the paleo-shoreline deposits [11,12]. Stratigraphically, this shale section is subdivided into three members on the basis of lithology, mineral composition and detailed well log correlation: the Evie Member, the Otter Park Member, and the Muskwa Formation, representing the Givetian to early Frasnian stages, respectively (~392 to 384 Ma) [21]. Although the age of the Canol Formation ranges from the late Givetian to middle Frasnian stages (383.2–367.7 Ma), the geologic age of the Horn River Formation is still disputed [19,22].

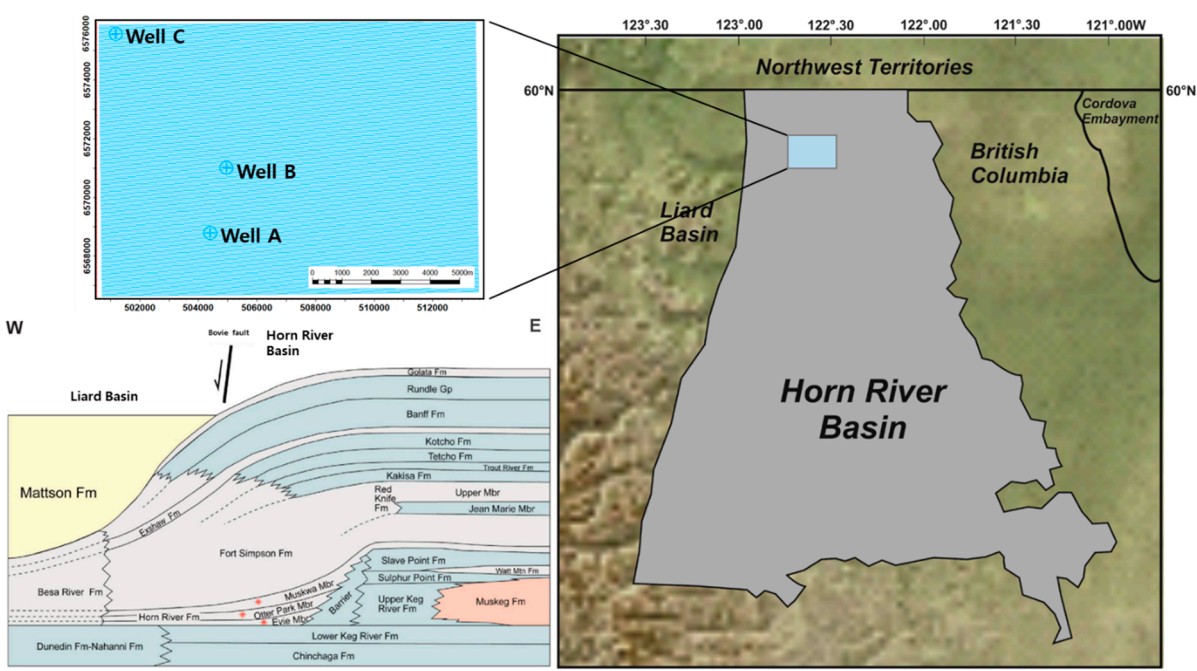

**Figure 1.** Well location in the Kiwigana Field of the Horn River Basin. Note the relationship between the Horn River Basin and the Liard Basin in terms of their location and stratigraphic cross section [15].

The Evie Member is calcareous mudstone that overlies the shallow marine carbonates of the Lower Keg River Formation [8,23]. It is up to 75 m thick near the Presqu'ile barrier and thins to less than 40 m to the west [23]. The well log response is characterized by a high gamma ray signature and high resistivity. The Evie Member has a high TOC

content [7,18,24]. The Otter Park Member is characterized by light gray to dark gray, pyritic, and calcareous to non-calcareous mudstone [7,11]. This member is thicker than the underlying Evie Member and the overlying Muskwa Formation and is up to 270 m thick in the south-eastern part of the Horn River Basin [23].

　　Organic content of the Otter Park Member is lower than that of the Evie Member and Muskwa Formation [7,18]. Siliciclastic and carbonate sediments of three members have been supplied from the southern part to the northern part [11,18]. The Otter Park Member becomes less calcareous towards the north and west, away from the distal parts of the basin [7,11]. The Muskwa Formation is comprised of a gray to black siliceous, pyritic, organic-rich shale, which overlies the Otter Park Member [7,23]. It varies in thickness from 50 to 90 m [17]. The organic carbon content of the Muskwa Formation is higher than that of the Otter Park Member, but lower than that of the Evie Member [7,23].

## 3. Materials and Methods

### 3.1. Data

　　The study area in the Kiwigana Field is located in the north-western part of the Horn River Basin, where seismic data, three logged wells, and one cored well are available (Figure 1). Wireline log data and core samples cover the sedimentary successions of the Muskwa Formation, Otter Park, and Evie Members of the Horn River Basin. Both conventional and advanced log data (for example, elemental capture spectroscopy (ECS)) were acquired at Well A. A TOC analysis was conducted on 145 rock samples using a LECO instrument. X-ray diffraction (XRD) analysis and thin section observations of 95 rock samples were performed by TerraTek in Calgary, Canada and at the KIGAM (Korea Institute of Geoscience and Mineral Resources in Daejeon, Korea) to obtain information on the biotic content and mineralogy.

### 3.2. Analytical Method

　　The cores were described in terms of lithological, sedimentological, and paleontological features in millimeter and centimeter scales. The geochemical data were integrated with the XRD (45 samples) and ECS data in order to classify the Horn River Formation in detail. This analysis resulted in nearly continuous (10 to 20 cm intervals) proxy records of the mineralogical composition of the core, including total clay, total carbonate, and QFM (quartz, feldspar, and mica).

　　The 38 samples were analyzed for porosity and permeability. Porosity and permeability were obtained from a tight rock analysis by TerraTek. Porosity was determined through bulk and grain volumes, obtained by the mercury immersion method and the helium pycnometry method [25]. Permeability was determined by analyzing pressure decay on a crushed sample.

　　Although the TOC content can be analyzed directly from geochemical data, it can also be derived from the wireline data. The TOC obtained through geochemical analysis is compared with results from the Schmoker density log method and the Passy $\Delta$logR method [26,27]. The Schmoker method assumes that the density of the formation depends on the presence or absence of low-density organic matter (1.0 g/cm$^3$). The $\Delta$logR method is applied to the sonic and resistivity log.

　　The brittleness index (BI) is a key parameter to determine how ductile or brittle a rock formation is, which is commonly calculated based on its mineral composition. This study calculates BI according to the following equation:

$$BI = quartz/quartz + carbonate + clay \qquad (1)$$

　　We interpreted 3D seismic data (12.8 × 9.2 km$^2$) and plotted a 3D grid of seismic data to analyze seismic facies and petrophysical modeling of the study area. This process enabled us to understand the property distribution and heterogeneity of the shale reservoir.

## 4. Results

### 4.1. Lithofacies Description and Mineral Components

Six lithofacies were identified and interpreted based on texture, bedding style, color, and mineral composition along the depth interval of 2307–2470 m. These include: faintly laminated siliceous mudstone (FLSM), homogeneous siliceous mudstone (HSM), laminated siliceous mudstone (LSM), laminated mixed mudstone (LMM), argillaceous mudstone (AM), and calcareous mudstone (CM). XRD and ECS log data are incorporated to quantify mineralogical abundance within each of the lithofacies (Table 1). Details of each of the facies and their characteristics are summarized in Figure 2.

The dominant facies are: FLSM: 39.9%, LMM: 33.7%, and HSM: 16.7%. The other three subordinate lithofacies are: LSM: 3.9%, AM: 1.9%, and CM: 3.9% (Figure 3). FLSM and HSM are characterized by a dark gray color, having higher QFM and TOC contents than the other lithofacies (Figure 3). The high QFM contents affect hydraulic fracturing, and the TOC content is related to gas volume [28,29]. These two organic-rich shale facies therefore tend to be brittle. Two other shale lithofacies, LMM and LSM, have thin or thick laminations and low TOC contents (Figure 3). AM contains the highest clay percentage and an average TOC content of 2.39 wt%. CM is a relatively carbonate-rich lithofacies within the studied core interval and possesses an average TOC content of 3.67 wt% (Figure 3).

**Table 1.** Mineralogical compositions of the six core facies. FLSM: faintly laminated siliceous mudstone. HSM: homogeneous laminated siliceous mudstone. LSM: laminated siliceous mudstone. LMM: laminated mixed mudstone. AM: argillaceous mudstone. CM: calcareous mudstone.

| QFM Contents (wt%) | | | | | | |
|---|---|---|---|---|---|---|
| Facies | FLSM | HSM | LSM | LMM | AM | CM |
| min | 27.3 | 37.0 | 55.2 | 34.0 | 37.3 | 1.7 |
| max | 88.7 | 87.6 | 84.6 | 70.0 | 45.7 | 37.4 |
| avg | 64.5 | 72.2 | 59.9 | 43.1 | 42.7 | 22.5 |
| **Carbonate Contents (wt%)** | | | | | | |
| Facies | FLSM | HSM | LSM | LMM | AM | CM |
| min | 0.6 | 0.9 | 8.1 | 0.6 | 0.6 | 51.7 |
| max | 38.6 | 48.6 | 24.8 | 54.4 | 12.6 | 85.1 |
| avg | 4.2 | 6.6 | 10.9 | 22.1 | 4.0 | 71.9 |
| **Clay Contents (wt%)** | | | | | | |
| Facies | FLSM | HSM | LSM | LMM | AM | CM |
| min | 65.2 | 8.3 | 15.4 | 55.4 | 45.1 | 0.3 |
| max | 3.5 | 43.8 | 33.9 | 13.4 | 62.7 | 10.9 |
| avg | 31.4 | 21.3 | 29.1 | 34.8 | 53.3 | 5.5 |

### 4.2. Sequence Stratigraphic Interpretation of Horn River Formation Succession

A previous study [12] suggested that the Horn River Formation formed in response to a combination of temporal and spatial cycles of base-level movement [30,31]. The key stratigraphic surfaces, which are correlated with stratigraphic discontinuities, were identified by changes in lithofacies, uranium logs, and Th/U ratios. Following this, gamma ray logs are correlated with the stratigraphic key surface of the existing depositional sequences [12]. Three depositional sequences of the Horn River Formation in the study area are as follows (Figure 4).

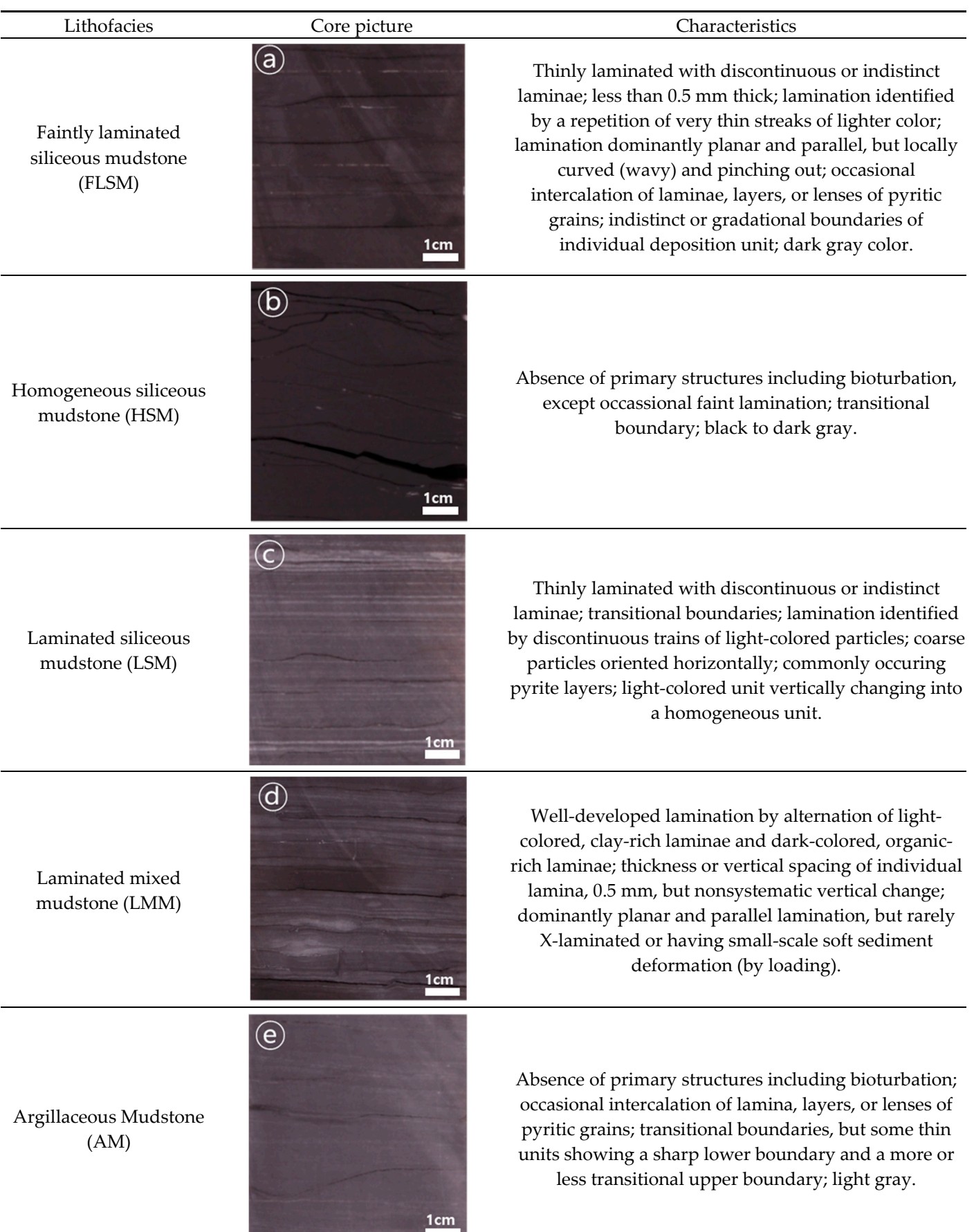

| Lithofacies | Core picture | Characteristics |
|---|---|---|
| Faintly laminated siliceous mudstone (FLSM) | | Thinly laminated with discontinuous or indistinct laminae; less than 0.5 mm thick; lamination identified by a repetition of very thin streaks of lighter color; lamination dominantly planar and parallel, but locally curved (wavy) and pinching out; occasional intercalation of laminae, layers, or lenses of pyritic grains; indistinct or gradational boundaries of individual deposition unit; dark gray color. |
| Homogeneous siliceous mudstone (HSM) | | Absence of primary structures including bioturbation, except occasional faint lamination; transitional boundary; black to dark gray. |
| Laminated siliceous mudstone (LSM) | | Thinly laminated with discontinuous or indistinct laminae; transitional boundaries; lamination identified by discontinuous trains of light-colored particles; coarse particles oriented horizontally; commonly occuring pyrite layers; light-colored unit vertically changing into a homogeneous unit. |
| Laminated mixed mudstone (LMM) | | Well-developed lamination by alternation of light-colored, clay-rich laminae and dark-colored, organic-rich laminae; thickness or vertical spacing of individual lamina, 0.5 mm, but nonsystematic vertical change; dominantly planar and parallel lamination, but rarely X-laminated or having small-scale soft sediment deformation (by loading). |
| Argillaceous Mudstone (AM) | | Absence of primary structures including bioturbation; occasional intercalation of lamina, layers, or lenses of pyritic grains; transitional boundaries, but some thin units showing a sharp lower boundary and a more or less transitional upper boundary; light gray. |

**Figure 2.** *Cont*.

| | | |
|---|---|---|
| Calcareous Mudstone (CM) | 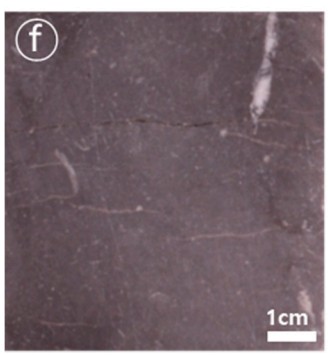 | Fine sand-sized bio-clasts randomly dispersed (disorganized) or faintly laminated; no distinct unit boundary; occasional stylolitic surface; mostly medium light gray. |

**Figure 2.** Description of classified core facies and their pictures. The core descriptions and mineralogical data span a depth range of 2307.6–2469.95 m. (**a**) Faintly laminated siliceous mudstone (FLSM), (**b**) Homogeneous siliceous mudstone (HSM), (**c**) Laminated siliceous mudstone (LSM), (**d**) Laminated mixed mudstone (LMM), (**e**) Argillaceous Mudstone (AM), (**f**) Calcareous Mudstone (CM) lithofacies.

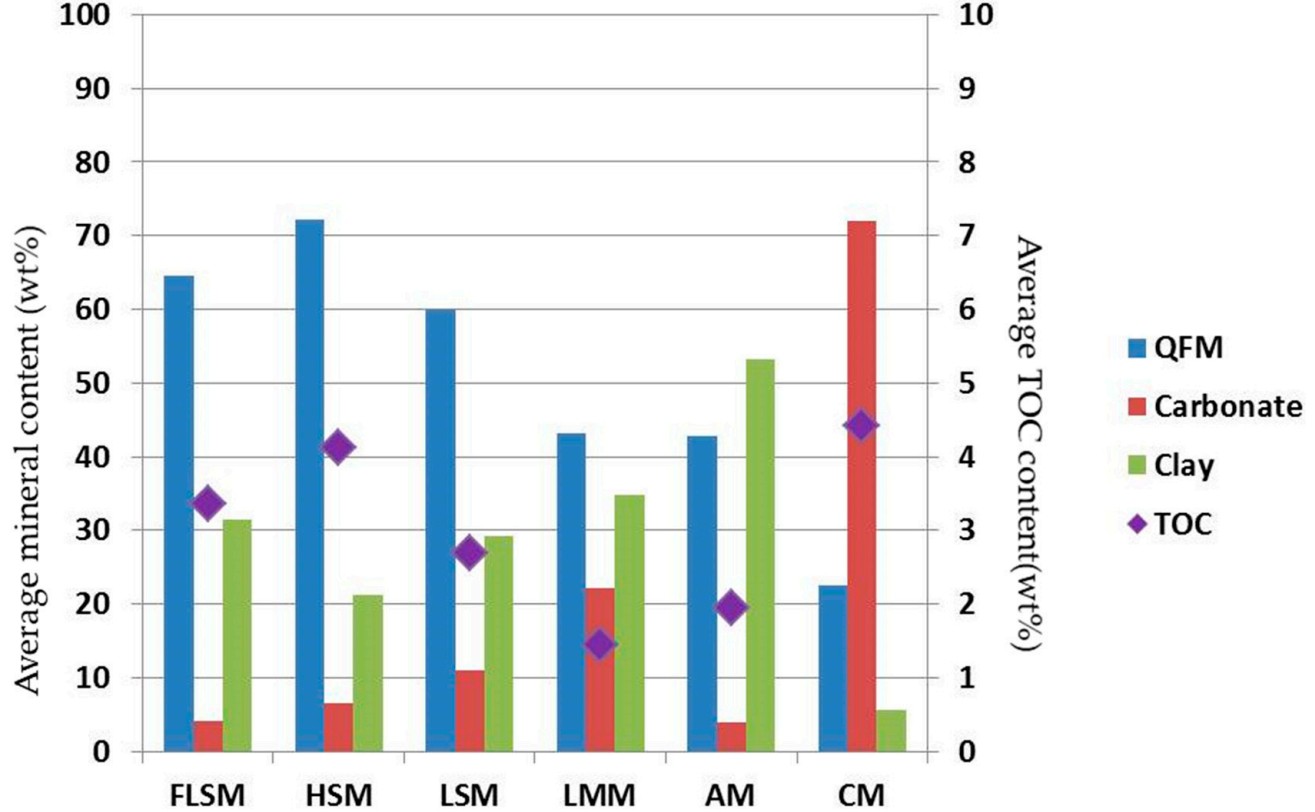

**Figure 3.** Mineralogical compositions of the six core facies. FLSM: faintly laminated siliceous mudstone. HSM: homogeneous laminated siliceous mudstone. LSM: laminated siliceous mudstone. LMM: laminated mixed mudstone. AM: argillaceous mudstone. CM: calcareous mudstone. QFM: quartz, feldspar, and mica. TOC: total organic carbon contents.

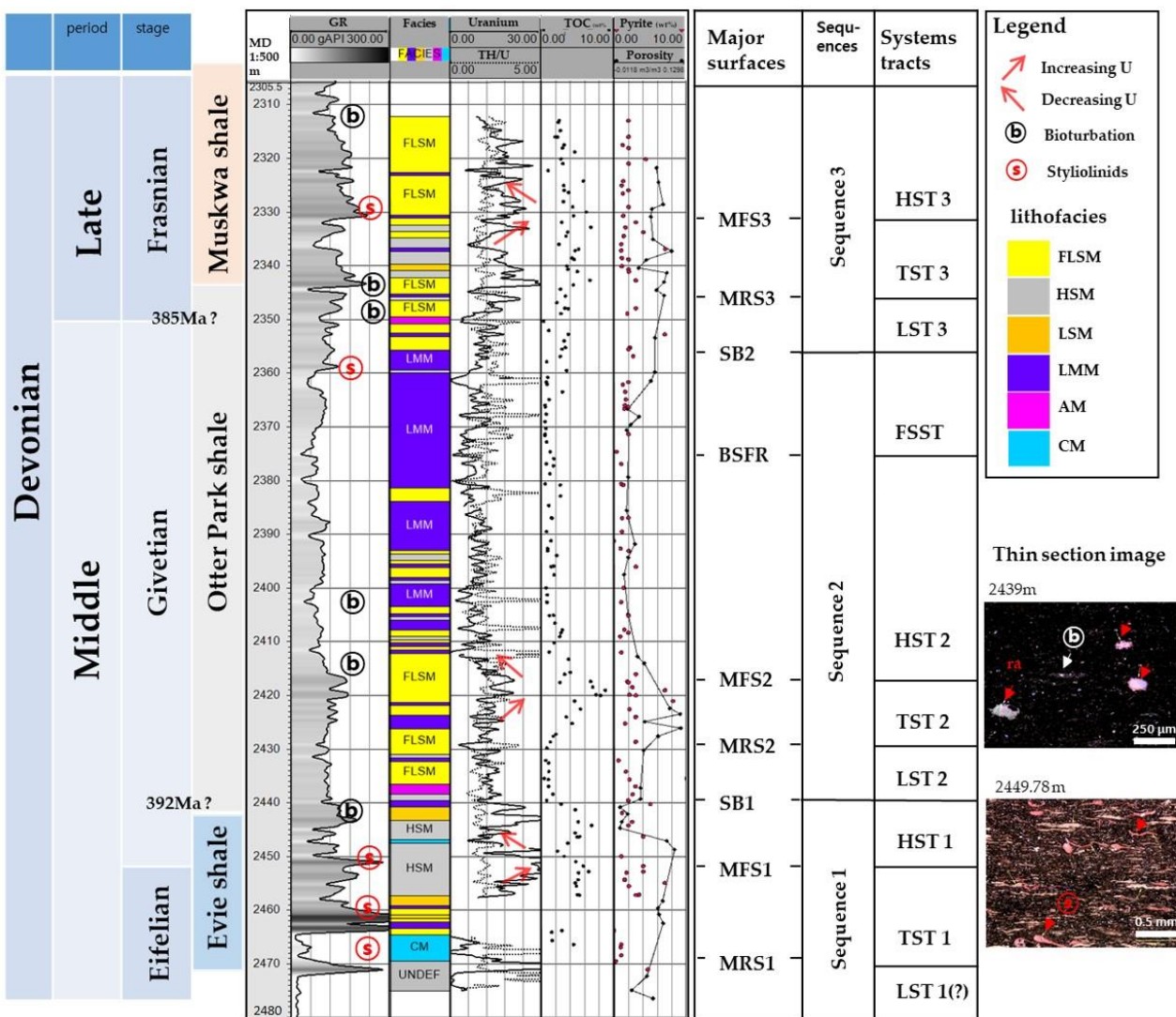

**Figure 4.** Sequence stratigraphic interpretation of Well A that is correlated with the depositional sequences of [12]. GR: natural gamma ray. Th/U and U: derived from the spectral gamma ray log. TOC points: geochemical analysis, Pyrite: XRD data, porosity, and core facies (see the legend for details).

Sequence 1: This is an incomplete sequence comprising of a highstand systems tract (HST) overlying the transgressive systems tract (TST). The highstand systems tract can be separated from the transgressive systems tract based on the cyclic pattern of the gamma ray log, representing a change from upward decrease to upward increase. The uranium logs are high in the upper-part, and there is a high Th/U ratio in the lower-part of this sequence. Styliolinids are occasionally observed in this sequence, which may indicate a more argillaceous nature, suggesting quiet offshore conditions [32]. Devonian Styliolinids indicate pelagic sedimentation [33]. The lithofacies in this sequence are comprised of two parts: upward coarsening (from LMM/HSM to LSM), and upward fining (from LMM to HSM facies) trends (Figure 4).

Sequence 2: This sequence comprises the whole depositional sequence. It is bounded below by sequence boundary one (SB1) and above by sequence boundary two (SB2). This sequence comprises the lowstand systems tract (LST), transgressive systems tract, the highstand systems tract, and the falling stage systems tract (FSST). It shows a general decrease in gamma and uranium log responses (Figure 4). The upper contact of this sequence is characterized by a dominant lithofacies change from LMM to FLSM. This sequence has low TOC (avg. 1.55 wt%), uranium (avg. 9.7 wt%), and total gamma ray

log responses (avg. 172 API). The lower part of this sequence shows subtle bioturbation (Figure 4). The uranium logs are much higher in the HSM and FLSM facies than in the LMM facies. The upward-decreasing uranium log pattern represents a retrogradational stacking pattern that contains a facies change from FLSM to LMM, whereas the upward-increasing uranium pattern represents a progradational stacking pattern that contains a facies change from LMM to FLSM/HSM.

Sequence 3: This sequence comprises the lowstand systems tract, transgressive systems tract, and the highstand systems tract. It shows a general decrease in gamma and uranium log responses in the HST and an upward increase in gamma and uranium log responses in the TST (Figure 4). The HST is generally dominated by the FLSM facies. The transgressive systems tract is intercalated with LMM, LSM in HSM, and the lower part with FLSM, whereas the lowstand systems tract is intercalated with AM in the FLSM facies (Figure 4). The transgressive systems tract represents high TOC and pyrite contents with high porosity. The abundance of pyrite influences brittleness, and therefore affects the completion quality and reservoir reserve estimates [34].

Lamination is suggested to be critical in reconstructing redox conditions [35]. The anaerobic to dysaerobic transition is characterized by faint laminations, whereas the anaerobic zone is suggested by well-developed laminations under pycnocline conditions [35]. The laminae are assumed to have formed in response to small-scale fluctuations within a single flow or depositional event, such as boundary layer bursts, sweep under currents, wave-oscillation currents, and the seasonal growth of planktonic or benthic organisms [36]. The continuity of the laminae decreases dramatically at 2356 m depth, accompanied by a change in the dominant facies from LMM to FLSM.

### 4.3. Structure Map of Sequence and Petrophysical Modeling

On a seismic profile, four lithostratigraphic units and three sequences are recognized based on distinct seismic horizons (Figure 5a). Well-to-seismic ties suggest that SB1 correlates to the top of the Otter Park horizon and SB2 correlates to the top of the Evie horizon. Figure 5b–d provides the sediment thickness distribution trends of three sequences. Each sequence has a distinctive thickness distribution pattern (Figure 5b–d). The thickness of Sequence 1 in the study area is relatively uniform, except in the south-western part where its thickness decreases significantly (over 10 ms). Stratal thickness of Sequence 2 gradually decreases from east to west.

The stochastic quantitative modeling of lithofacies in three dimensions reveals detailed spatial variations and helps to figure out the depositional environments and their geological characteristics [37]. The first step in petrophysical modeling is to build the three-dimensional lithofacies model. Six lithofacies of three wells are established, upscaled and used as hard data for facies modeling (Figure 5e). The lithofacies modeling of the Horn River Formation shows differing distribution patterns: Evie dominates HSM facies, Otter Park dominates LMM facies, and Muskwa represents FLSM dominant lithofacies. These facies modeling results are comparable with lithofacies correlation in that they are generally diachronous surfaces.

Derived TOC and a brittleness index for petrophysical modeling would verify the effectiveness of the lithofacies model. The TOC log was calculated by the ΔlogR method and the BI was calculated from XRD data. The derived TOC data is relevant to the vertical variation of the total organic carbon, which represents the distribution of geological characteristics of the Horn River Formation reservoir. Spatial distribution of the TOC and brittleness index was simulated using the sequential Gaussian simulation method to obtain a TOC and brittleness index map (Figure 5f,g). We observed that the TOC and BI distribution maps show similar trends. A petrophysical evaluation of the Horn River Formation indicates that a higher TOC content results in an increased BI (Figure 6).

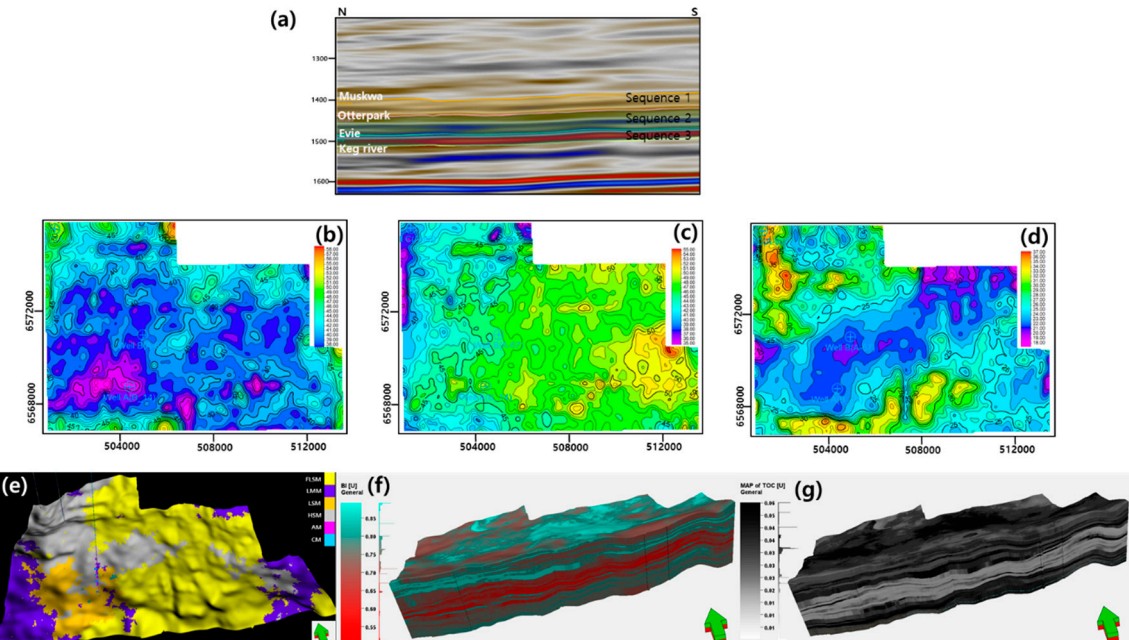

**Figure 5.** Seismic data interpretation results: (**a**) original seismic data, (**b**) time thickness map of Muskwa, (**c**) time thickness map of Otter Park, (**d**) time thickness map of Evie, (**e**) lithofacies modeling map of Muskwa, (**f**) petrophysical modeling map of brittleness index, and (**g**) petrophysical modeling map of the TOC.

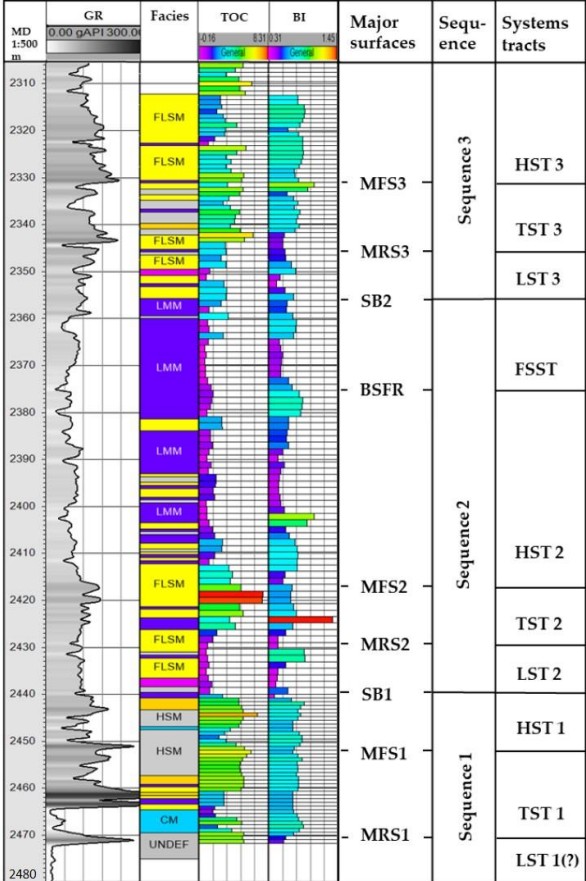

**Figure 6.** Petrophysical parameters of Well A. GR: core gamma ray. Facies: core facies. TOC: upscaled derived TOC from the Passy method. BI: upscaled brittleness index.

## 5. Discussion

### 5.1. Key Stratigraphic Surfaces of the Horn River Formation

The recognition of abrupt stratigraphic discontinuities is especially important in sedimentological reconstructions and sequence stratigraphic analyses. The strata discontinuity has been determined through spectral gamma ray logs or their ratios [38–42], because spectral gamma rays contain more detailed information on sedimentary rocks [43]. Th/U against the U-based method and lithofacies changes were applied to detect abrupt breaks and stacking patterns of the Horn River Formation. Several abrupt stratigraphic breaks were identified in the study intervals, and such features were especially prominent through spectral gamma rays and their ratios (Figure 4). The maximum flooding surface (MFS) shows higher uranium peaks, and the maximum regressive surface (MRS) represents a high Th/U ratio with a lower uranium log. The sequence boundaries are characterized by consistent Th/U ratios and uranium logs. Two sequence boundaries were identified in the Horn River Formation. Sequence boundaries within the shale deposit tend to be very subtle, but they are much more extensive. These key stratigraphic surfaces are correlated based on gamma rays and match with previous results of Ayranci (2018) [12]. SB1 shows a significant drop in the uranium log and TOC contents, and SB2 represents the change of the dominant lithofacies from LMM to FLSM, which suggests a change in sediment type or depositional environment. The maximum flooding surface is associated with abundant organic matter, which gives rise to a high uranium content of organic-rich shale [44–46]. Oxidized continental sediments have higher Th/U ratios than non-oxidized marine sediments. Time equivalent portions of the Horn River Formation reveal different lithofacies, indicative of facies transition within the system tract at the time of deposition.

### 5.2. Porosity, Permeability, and TOC

Porosity, permeability, and TOC parameters are largely dependent on lithofacies (Figure 7). Porosity and total organic carbon are both important to define the geological sweet spot [2,47]. The Horn River Formation shows considerable variation in its lithofacies and mineral contents (Table 1), and in petrophysical properties such as porosity, permeability, and TOC. The HSM and FLSM lithofacies show high porosity and permeability with high QFM contents (>64%) (Table 1). The lowest porosities are related to LMM and CM with high carbonate contents that are tightly compacted or cemented with carbonates (Figure 7). The lithofacies have a strong relationship with porosity, permeability, and TOC content, and their correlation coefficient is relatively high. In addition, the trend of higher quartz concentration goes along with higher TOC content (Figure 3). Additionally, TOC trends of transgressive systems are overall of higher value than those of the highstand and lowstand systems tracts (Figure 8).

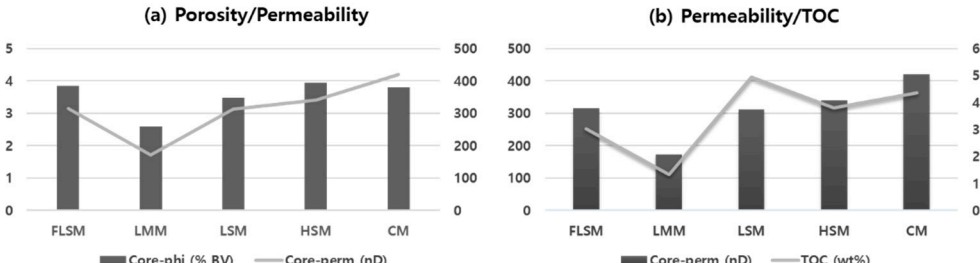

**Figure 7.** The relationship between lithofacies and (**a**) porosity and permeability, and (**b**) permeability and TOC content, measured by core analysis data of the Horn River Formation.

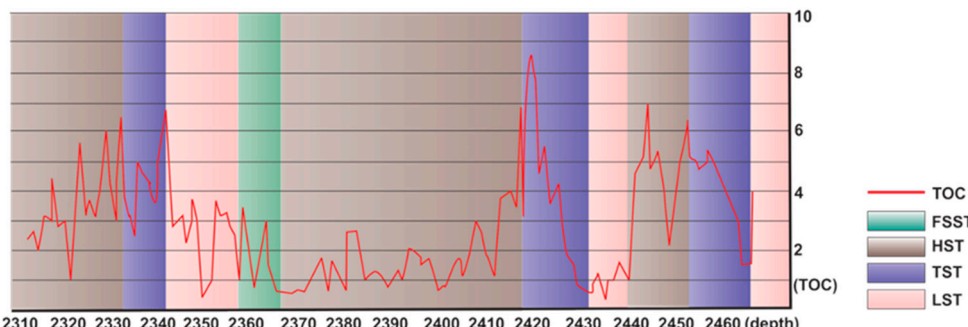

**Figure 8.** The relationship between systems tract and TOC content.

Previous studies have tried to figure out how to expand from the change of water depth and lithofacies type to the environmental change of redox conditions, which proves that the change in TOC may be related to reduction conditions [48,49].

The TOC content is important for reconstructing the redox conditions and reservoir quality [24,50]. The TOC content represents the weight percentage of carbon in the rock, and therefore depends on the productivity and preservation of organic matter. The sequence stratigraphic interpretation indicates that organic carbon content generally increases toward the maximum flooding surface [3], and then decreases stepwise (Figure 4). However, the distribution of organic matter is not significantly affected by the systems tracts. According to the sequence stratigraphy analysis of the Horn River Formation, the transgressive systems tract shows an overall increase in TOC content. However, the other systems tract does not have distinctive patterns (Figure 8). According to the correlation with the depositional sequences of earlier studies of Ayranci (2018) [12], these results indicate that the TOC difference is influenced by other factors. As such, we consider the redox condition from TOC variation.

### 5.3. Importance of Paleo Redox Condition

The effect of the sediment accumulation rate on organic matter preservation and TOC is strongly dependent on the prevailing paleo redox condition. Most of the black shales are accumulated beneath dysoxic or anoxic water, and organic-rich shales accumulate as a result of enhanced preservation of organic matter in anoxic environments [51]. Dysoxic–anoxic facies show either a largely independent or negative relationship between TOC content and sediment accumulation rate [52–54]. According to the sequence stratigraphy analysis in the Horn River Formation, the transgressive systems tract shows increasing TOC and uranium contents, while the highstand systems tract shows decreasing TOC and uranium contents (Figures 4 and 8). These features suggest that transgressive shales are organic-rich deposits and regressive shales are organically lean [55,56]. The uranium concentration is high through the transgressive systems tract, which implies an anoxic condition [24]. Previous work interpreted that the Evie Member, the middle Otter Park Member, and the Muskwa Formation tend to have anoxic conditions, whereas the lower/upper Otter Park Members tend to have suboxic conditions, based on the redox-sensitive trace element analysis [24].

Th/U ratios have been used as a chemo-stratigraphic proxy to determine whether the sediments originated from marine or continental environments [38,52,57,58]. Oxidized continental sediments have higher Th/U ratios than non-oxidized marine sediments. Redox conditions in the depositional sequence are classified as anoxic Th/U (<2) by Wignal and Twitchett (1996) [59]. However, suboxic to anoxic conditions are confirmed by Th/U < 0.8 [60]. Since most of the systems tracts show that Th/U < 2, the Horn River Formation was primarily deposited in an anoxic quiescent basin plain (Figure 4). Only part of HST2, FSST, and LST show high values, i.e., higher than 2 in the Th/U ratio.

## 6. Conclusions

The Horn River Formation has been interpreted as three depositional sequences based on sedimentological, geochemical, and petrophysical features. A sequence stratigraphic analysis leads us to figure out the redox condition, energy regime, and petrophysical parameters, which can be related to lithofacies characteristics that are relevant to the geological sweet spot. Individual sequences are commonly separated by distinctive relatively low or high TOC content. The relationship between TOC trends and the systems tract is insignificant whereas the redox condition is relevant.

The lithofacies variables, through existing depositional sequences, are comparable to the porosity, permeability, and TOC content. A change in lithofacies represents a major change in sedimentological conditions and may or may not be part of the sequence. However, geological characterization of shale lithofacies, in terms of sequence stratigraphy combined with integration of well logs and geochemical data, would be useful to find the geologically sweeter reservoir.

**Author Contributions:** J.W. designed research; C.W.R., S.H.Y. and J.C. performed research and analyzed data. All authors have read and agreed to the published version of the manuscript.

**Funding:** This work was supported by the Basic Science Research Program through the National Research Foundation of Korea (NRF) funded by the Ministry of Education (2021R1I1A1A01052612, 2020R1F1A1075624).

**Institutional Review Board Statement:** Not applicable.

**Informed Consent Statement:** Not applicable.

**Data Availability Statement:** Not applicable.

**Acknowledgments:** The authors thank Schlumberger and CGG for their support through a university licensing program of Petrel 2018 and HRS 10, respectively.

**Conflicts of Interest:** The authors declare that they have no conflict of interest.

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
