# Peer review of "Verification and Application of Sequence Stratigraphy to Reservoir Characterization of Horn River Basin, Canada"

_minerals, doi:10.3390/min12060776_

Round 1

Reviewer 1 Report

Dear Authors,

your contribution is interesting and gives useful info on sequence boundary recognition within shales sequences (e.g. geochemical proxies). However, it could be improved with relevant references (e.g. Massimo Zecchin, Octavian Catuneanu, 2015 - High-resolution sequence stratigraphy of clastic shelves III: Applications to reservoir geology. Marine and Petroleum Geology 62 161-175 and references therein). Moderate revision of English language is requested and figures need also to be improved. Please, find an attached file with some suggestions.

Author Response

The authors appreciate the reviewer’s concern regarding the sequence boundary recognition within the shales sequences. I read the paper the reviewer recommended and update the sequence stratigraphic analysis. Also we have adopted all the suggestions and the highlighted the changes in the revised manuscript.

Reviewer 2 Report

Some of my major concerns:

I have major concerns regarding facies analysis. The authors list six facies, but they only provide names. There is no detailed sedimentological description which is also crucial for sequence stratigraphy. Also, they provide one small figure with no close-up images or thin sections, also no scale bar! Almost all figures look very similar, so on what basis these facies were differentiated? This becomes a clearer issue when authors describe these facies and their TOC content: For example, in lines 147-148 and 151-152, they combine two facies together twice without showing how those two are different from each other. It makes it very difficult to follow. Moreover, there is an interpretation following these lines which should be separated from results, and TOC data under the section of “lithology and mineral component”. So, in a nutshell, lithofacies descriptions require major revisions, I suggest authors provide detailed analysis of lithology and sedimentary structures along with representative figures.

Given that sedimentological analysis is lacking substantial detail, sequence stratigraphy cannot be justified. The authors state that they used gamma-ray and lithofacies changes, but there is no explanation about how they constructed such a framework. Gamma-ray has been used to identify T-R cycles, but 4-stage systems tracts require facies. Authors should provide a detailed sedimentological analysis and support their sequence stratigraphy using these analyses, not based on the presence of abundant Styliolinids, etc. I am very well aware of the Horn River Basin sequence stratigraphy, and their sequence stratigraphic framework does not match with other studies. Some major surfaces are off by at least 20 meters! The reasons for this should be clearly addressed in the text.

There are some data discussions in the “discussion” section, but no information in the methodology or results sections. For example, pyrite content and bioturbation. These have been well documented in the Horn River literature, but I don’t understand how authors make interpretations without documenting those first. Yes, I do see a pyrite column in Fig. 4, but I don’t understand what that range means (0-10) or why it is not mentioned in the results with some figures. Bioturbation is completely missing, no information.

Minor concerns:

Missing references throughout the text. This is particularly an issue in the geological overview as the text reads like the authors’ results.

XRD analysis results are given in the methodology but should be in the results, after describing facies.

Facies analysis only comes from one core, not from others, although authors state that they log some other wells.

I also would like to emphasize that the authors use changes in the bioturbation between lines 191-192, but this has been published before, and since they don’t mention it in their methodology, it is not their results.

Line 202-204: Authors describe pyrite content here. Pyrite also is well documented in other papers on the horn river, but authors do not show any evidence of the presence of pyrite, but they mention it in their sequence stratigraphy. This is controversial.

Lines 205-207: Lamination or faint lamination is not related to the oxygen content in the water column. It is related to energy variations.

The authors state that they have seismic data; however, there is no evidence of a seismic image or even where they collected this data.

Lines 243-254: this entire paragraph should be removed. There is no discussion regarding the authors’ results, but there is only some textbook information related to overall sequence stratigraphy.

Horn River Formation, not Shale

Line 60: Figure 1 shows the study area, not a stratigraphic column showing age-equivalent units.

Line 79-81: add a reference.

Author Response

The author sincerely appreciates the reviewer’s evaluation of the manuscript and provision of constructive comments and recommendations. The sedimentological description has been added in Figure 2. I have a thin-section analysis, which is not used in the manuscript because sedimentary study is not the goal. However, I have added a thin-section image in Figure 4 of bioturbation and styliolinoids.

As I have made a fundamental error with the sequence stratigraphic analysis because of limited data available, I have now revised the results of the sequence stratigraphy analysis. In this study, three logged wells, and one cored well are available. Finding the key surface was done in the cored well and it was based on the spectral gamma rays and their ratio. Then I conducted the correlation using gamma rays through the Ayranchi (2018) paper. I welcome any further feedback you may have.

The Discussion part was revised based on your critical comments and the reviewer 3 opinion. The revision details can be found in the revised manuscript.

Reviewer 3 Report

This study attempts to use a variety of methods to study shale facies and its related geological characteristics.

The main recommendations are: 

1. It is suggested to systematically analyze the distribution law of shale facies in the sequence framework.
2. Deeply analyze the differences of porosity, permeability and TOC of different shale facies.
3. Comprehensively analyze the formation environment from the sequence framework and system tract of the lithofacies, combined with TOC, typical logging Th / u and other data, analyze the discrimination of sea-level rise and fall and reduction conditions, and explore the development environment of shale lithofacies.
That is, this paper should focus on the formation environment of "shale facies".

The detailed suggestions are as follows:

  1. It is necessary to further sort out the ideas and logic of the preparation of the abstract.
  2. There have been many studies on the relationship between lithofacies and reservoir performance. The "poor study" mentioned in this paper may specifically refer to the study area in Introdution.
  3.  Introdution. The first paragraph needs to further deepen the discussion on the significance of this study, and the summary of previous research is not enough; Combined paragraphs 2, 4, 5 and the last sentence of the first paragraph and put it at the end of the Introduction.
  4. 3. Previous studies of the Horn River Shales: This paragraph can be placed in the introduction. The understanding and deficiency of previous studies belong to the research significance.
  5. Table1 should appear in the results. In addition, the number of samples should be indicated.
  6. P4, line 149. There is no need for such a discussion here. Dong et al. (2015) conducted systematic research on lithofacies. Please give the criteria for dividing lithofacies and supplement the description of each lithofacies.
  7. P5. Supplement the photo number in Figure2 and annotate each lithofacies to describe its main characteristics.
  8. P6. Figure 4 needs to be redrawn instead of simply putting the two figures together. XRD can be extended to this figure, which is clearer than figure 3.
  9. P7. Line213-231.  How to verify the rationality of the method and the credibility of the results? What is the relationship between Figure 6 and the core content of the full text? This part of the work seems to have no relevance to the whole article, or the relevance is not enough.
  10. P7. Line233. Figure5
  11. P8. In Fig. 6, the sequence division should be added, and the relevant information such as strata and members should be supplemented to make the map complete. 
  12. P8.line 242-269. How to integrate the previous research with the results of this paper, rather than separate and say different things.
  13. P9. Line280. Porosity, permeability and other data can also be placed on the histogram in Fig. 6,, or the cross plot of porosity and permeability can be used to show the differences of porosity and permeability of different lithofacies. The number of samples is not given in the figure 7, and the ordinate has no unit, and the format is not standardized.
  14. P9. Line 283. TOC is higher in highstand and transgressive system tracts on the body, and it is affected by the rise and fall of system tracts or horizontal plane on the body, while the reduction condition is inference and there is no direct evidence. Therefore, more "supporting evidence" is needed for the discussion here. For example, some previous studies know how to extend from the change of water depth and lithofacies type to the environmental change of "reduction conditions", which proves that the change of TOC may be related to the reduction conditions.
  15. P10. Line302. Fig. 9 is redundant, and the relevant laws can be shown in Fig. 4 and Fig. 6. And Fig. 8 is similar to Fig. 9.
  16. P11. Line 320-322. The characteristics of the study area need to be deeply dissected and analyzed at 6.3. Importance of paleo redox condition.

Author Response

The authors are very grateful to the reviewer for providing constructive comments and recommendations. The manuscript has been revised to address the reviewer’s comments, which are uploaded alongside this document.

Round 2

Reviewer 3 Report

The author has revised the manuscript as suggested by the reviewer. The manuscript can be accepted. 

It is necessary to further strengthen the discussion on the relationship between sequence stratigraphy and paleoredox environment

Author Response

Response:

The authors are very grateful to the reviewer for providing constructive comments and recommendations. The manuscript has been revised to address the reviewer’s comments.

We admit that there is a limit to revealing the relationship between sequence stratigraphy and the paleoredox environment. This can be supplemented by previous work. They support the redox conditions for the formation unit.

s.
